# Association of the Metabolic Score Using Baseline FDG-PET/CT and dNLR with Immunotherapy Outcomes in Advanced NSCLC Patients Treated with First-Line Pembrolizumab

**DOI:** 10.3390/cancers12082234

**Published:** 2020-08-10

**Authors:** Romain-David Seban, Jean-Baptiste Assié, Etienne Giroux-Leprieur, Marie-Ange Massiani, Michael Soussan, Gérald Bonardel, Christos Chouaid, Margot Playe, Lucas Goldfarb, Boris Duchemann, Laura Mezquita, Nicolas Girard, Laurence Champion

**Affiliations:** 1Department of Nuclear Medicine, Institut Curie, 92210 Saint-Cloud, France; laurence.champion@curie.fr; 2Department of Pneumology, Paris-Est University, Centre Hospitalier Inter-Communal de Créteil, Inserm U955, UPEC, IMRB, équipe CEpiA, 94010 Créteil, France; jbaptiste.assie@gmail.com (J.-B.A.); christos.chouaid@chicreteil.fr (C.C.); 3Inserm, Centre de Recherche des Cordeliers, Sorbonne University, Functionnal Genomics of Solid Tumors Laboratory, F-75006 Paris, France; 4Department of Respiratory Diseases and Thoracic Oncology, APHP, Hôpital Ambroise Paré, 92100 Boulogne-Billancourt, France; etienne.giroux-leprieur@aphp.fr; 5Department of Medical Oncology, Institut Curie, 92210 Saint-Cloud, France; marieange.massiani@curie.fr; 6Department of Nuclear Medicine, Paris 13 University, APHP, Hôpital Avicenne, 93000 Bobigny, France; michael.soussan@aphp.fr (M.S.); margot.playe@gmail.com (M.P.); lucas.goldfarb@gmail.com (L.G.); 7Department of Nuclear Medicine, Centre Cardiologique du Nord, 93200 Saint-Denis, France; gerald.bonardel@gmail.com; 8Department of Medical Oncology, Paris 13 University, APHP, Hôpital Avicenne, 93000 Bobigny, France; boris.duchemann@aphp.fr; 9Department of Medical Oncology, Hospital Clínic, Laboratory of Translational Genomics and Target Therapeutics in Solid Tumors, IDIBAPS, 08036 Barcelona, Spain; LMEZQUITA@clinic.cat; 10Institut du Thorax Curie Montsouris, Institut Curie, F-75006 Paris, France; nicolas.girard2@curie.fr

**Keywords:** FDG-PET, NSCLC, first-line immunotherapy, derived neutrophils to lymphocytes ratio, total metabolic tumor volume, prognosis

## Abstract

*Background*: We aimed to assess the clinical utility of a previously published score combining the total metabolic tumor volume (TMTV) on baseline FDG-PET/CT and pretreatment derived from the neutrophils to lymphocytes ratio (dNLR) for prognostication in NSCLC patients undergoing first-line immunotherapy (IT). *Methods*: In this multicenter retrospective study, 63 advanced NSCLC patients with a PD-L1 tumor proportion score (TPS) ≥50%, who underwent FDG-PET/CT before first-line IT, treated from January 2017 to September 2019, were enrolled. Associations between this score and the progression-free survival (PFS), overall survival (OS), disease control rate (DCR), and overall response rate (ORR) were evaluated. *Results*: The median (m) PFS and mOS were 7.7 (95% CI 4.9–10.6) and 12.1 (8.6–15.6) months, respectively, and DCR and ORR were 65% and 58%, respectively. mOS was 17.9 months (14.6 not reached) for the good group versus 13.8 (95%CI 8.4–18.9) and 6.6 (CI 2.0–11.2) months for the intermediate and poor groups, respectively. mPFS was 15.1 (95%CI 12.1–20.0) months for the good group versus 5.2 (1.9–8.5) and 1.9 (95%CI 1.3–2.5) months for the intermediate and poor groups, respectively. The poor prognosis group was associated with DCR and ORR (*p* < 0.05). *Conclusions*: The metabolic score combining TMTV on the baseline FDG-PET/CT scan and pretreatment dNLR was associated with the survival and response in a cohort of advanced NSCLC patients with ≥50% PD-L1 receiving frontline IT.

## 1. Introduction

Immune checkpoint inhibitors (ICIs) have become standard-of-care for the treatment of patients with previously untreated advanced non-small cell lung carcinoma (NSCLC) [1,2]. The programmed cell death-1 (PD-1) inhibitor pembrolizumab is approved as monotherapy for newly diagnosed advanced NSCLC patients with a programmed cell death-ligand 1 (PD-L1) tumor proportion score (TPS) of at least 50%, and without a targetable oncogene such as ALK and EGFR [3,4], improving the clinical outcomes compared with the platinum-based chemotherapy (CT) [5]. However, even within the PD-L1-positive stratum [6], only a fraction of patients experience such therapeutic success. For example, in the pembrolizumab arm of the KEYNOTE-024 trial, the response rate was 45% [1]. The early identification of biomarkers for patients unlikely to benefit from first-line pembrolizumab is therefore a crucial step in selecting appropriate candidates [7,8].

Beyond the PD-L1 expression, easily measurable and accessible biomarkers for the prediction of outcomes are needed. Initial investigations, including a complete blood count and a 18F-Fluorodeoxyglucose positron emission tomography (FDG-PET), provide both diagnostic and staging information [9,10,11], but could also be predictive or prognostic [12,13,14,15,16,17,18]. On the basis of these two tests, certain parameters are beneficial and meaningful, particularly the derived neutrophils to lymphocytes ratio (dNLR) and the total metabolic tumor volume (TMTV) on FDG-PET [19].

On the one hand, the link between cancer, inflammation, and immunosuppression is well-recognized [20]. The accumulation of pro-inflammatory factors, such as dNLR, leads to immunosuppression, which is associated with cancer progression. An elevated dNLR has been reported to be associated with poor outcomes in some malignant tumors [21,22,23], including NSCLC, and its prognostic value for ICIs has been widely studied and demonstrated [24,25]. On the other hand, baseline FDG-PET imaging can provide relevant biomarkers in patients treated with ICIs [19]. In the foreground, TMTV, reflecting the whole tumor burden (TB), has value in predicting whether patients treated with anti-PD1 will respond to therapy, and whether they will die of melanoma [26,27] or NSCLC [19,28].

We previously developed a metabolic score, combining both TMTV on baseline FDG PET/CT scan and pre-treatment dNLR, which was correlated with IT outcomes, in two independent retrospective cohorts of pretreated NSCLC patients [19,29]. In the present study, we thus aimed to investigate the clinical utility of this metabolic score on the prognostication of clinical outcomes, in a multicenter cohort of untreated patients with advanced NSCLC with PD-L1 TPS of at least 50%, undergoing first-line pembrolizumab.

## 2. Results

### 2.1. Patient Characteristics

The baseline clinicopathologic characteristics, as well as the biological and PET imaging parameters of the 63 patients are summarized in Table 1. For the correlation between the biomarkers, refer to the Appendix A. Thirty-eight (60%) patients were male. The median age of the patients was 65 (range, 37–86) years old and 80% had non-squamous NSCLC (mainly adenocarcinoma, *n* = 49). Of the patients, 44 (70%) and 19 (30%) patients were high (TPS: 50–89%) and very high (TPS: 90–100%) PD-L1 expressors, respectively. Twelve patients were treated with corticosteroids (>10 mg of prednisone-equivalent dose) at baseline (Appendix A). There were no significant differences in patient characteristics between patients with or without corticosteroids (Appendix A), but also between high and very high PD-L1 expressors (Appendix A). The squamous cell carcinomas had a higher FDG uptake than the non-squamous cell carcinomas (Appendix A). Among the 39 (62%) patients with available genomic testing, molecular alterations were detected in 22 patients (including 17 patients with an activating KRAS mutation). A majority of patients had a blood test during the week preceding the first injection of pembrolizumab (median time 7.0 days), and the biological data were fully available, with the exception of the LDH levels, which were evaluable in a total of 40 patients (63%).

### 2.2. Correlation between Biomarkers

We found that TMTV and tumor SUVmax or dNLR did not correlate significantly with each other. We also found significant correlations between TMTV and age (rank (rho) = −0.29; *p* = 0.02), and LDH (rank (rho) = 0.41; *p* < 0.01). Furthermore, the tumor SUVmax was not correlated with any clinical or biological variables.

### 2.3. Metabolic Score and Correlation with Patient’s Outcomes

According to the metabolic score, 25 (40%) patients were in the good prognosis group, 27 (43%) in the intermediate prognosis group, and 11 (17%) in the poor prognosis group (Appendix A).

### 2.4. Survival: Progression-Free Survival (PFS) and Overall Survival (OS)

After a median follow-up of 13.4 (95%CI 9.0–17.9) months, 37 (59%) and 18 (29%) patients experienced progression and death, respectively. The median PFS and OS were 7.7 (95%CI 4.9–10.6) months and 12.1 (95%CI 8.6–15.6) months, respectively.

Both a high dNLR and high TMTV were significantly correlated with a poor PFS and OS (*p*-values < 0.05) in the univariate analysis (Table 2, Appendix A). High dNLR remained the only independent statistically significant parameter associated with PFS (HR 2.0, 95%CI 1.1–4.0) and OS (HR 3.4, 95%CI 1.3–8.8) in the multivariate analysis (Table 2).

Using the metabolic score, the median PFS was 13.8 (95%CI 8.4–18.9) months for the good prognosis group versus 6.6 (95%CI 2.0–11.2) months for the intermediate prognosis group, versus 1.9 (95%CI 1.3–2.5) months for the poor prognosis group (*p* = 0.01; Figure 1 and Table 3). The median OS for the good, intermediate, and poor metabolic score was 17.9 (95%CI 14.6-not-reached), 15.1 (95%CI 12.1–20.0), and 5.2 (95%CI 1.9–8.5) months, respectively (*p* < 0.01; Figure 2 and Table 3).

### 2.5. Response: DCR and ORR

A clinical benefit was observed in 65% of patients according to DCR, and 58% of patients according to ORR (2 with CR, 35 with PR, and 4 with SD). In the logistic regression and according to the DCR classification, low TMTV (≤75 cm^3^) and low dNLR (≤3) were statistically associated with a clinical benefit (odds ratio (OR) 3.8, 95%CI 1.2–11.6 and OR 4.9, 95%CI 1.5–16.3, respectively; Table 4). In the logistic regression and according to the ORR classification, low LDH (≤ULN) and low dNLR (≤3) were statistically associated with a clinical benefit (odds ratio (OR) 5.8, 95%CI 1.4–24.7 and OR 3.3, 95%CI 1.1–10.5, respectively; Table 4). Notwithstanding the consequent odds ratio and indicative confidence interval, TMTV failed to reach the statistically significant threshold, namely: TMTV ≤ 75 cm^3^ with an OR of 2.5 (0.9–7.0) and *p* = 0.09.

Correlations were also observed between the metabolic score and clinical benefit (Table 5 and Appendix A). The poor prognosis group was associated with the DCR and ORR (*p* < 0.01 and *p* = 0.04, respectively). The poor and intermediate prognosis groups were associated with progressive disease (non-DCR) as the best overall response to pembrolizumab (OR 9.8, 95% CI, 1.9–31.9 and OR 5.6, 95% CI, 1.3–23.4, respectively).

## 3. Discussion

Pre-treatment with TMTV and dNLR have a significant value in prognosticating the response and survival in advanced NSCLC patients treated with first-line pembrolizumab. Moreover, this study confirms the trend that the combination of these two factors allows for a more accurate prognostication for the clinical outcomes than when used alone. In addition, the metabolic score stratifies the population in three different prognostic groups. Interestingly, our score showed that a high-risk group of patients (TMTV > 75 cm^3^ plus dNLR > 3) encompassed 17% of the population with a median PFS of 1.9 months and a median OS of 5.2 months, and were more likely associated with primary resistance to first-line IT.

Our score stratified NSCLC patients under anti-PD-(L)1 inhibitors according to the response and survival outcomes, regardless of the number of prior therapy lines [19]. However, we recognized the importance of assessing such a scoring approach in an independent and more homogeneous population of NSCLC patients receiving first-line immunotherapy [30]. It is in such a framework that we tested the clinical relevance of this previously developed metabolic score so as to prognosticate the clinical outcomes in a new and multicenter cohort of patients with advanced NSCLC with a PD-L1 TPS of at least 50%, treated with first-line pembrolizumab. This scoring approach is an innovative way to select appropriate candidates who would not benefit from first-line pembrolizumab monotherapy in advanced NSCLC patients. The characteristics in the score are readily available to clinicians, making it calculable to provide personalized estimates of the risk of first-line pembrolizumab failure. dNLR can be obtained rapidly from the initial complete blood counts, and TMTV can be easily extracted by nuclear medicine specialists from baseline FDG-PET imaging using the standardized and reproducible method proposed.

Several studies confirm the strong relationship between systemic cancer-associated inflammation or metabolic TB, and poor outcomes in NSCLC patients receiving ICIs (Appendix A). Systemic inflammation is associated with defective myelopoiesis, which can be captured by the neutrophil to lymphocyte ratio (NLR) and the derived NLR (dNLR) [31]. Another score, including the total lesion glycolysis (TLG) on FDG-PET and NLR, can predict the survival outcomes in advanced NSCLC treated with anti-PD-(L)1 immunotherapy [28]. For instance, from 25 advanced NSCLC patients mainly treated in a second-line setting, Castello et al. developed an immune-metabolic-prognostic index (IMPI), combining the NLR and TLG extracted from the FDG-PET scan, both at the first restaging after three or four cycles, depending on the ICI received. Recently, the same authors have confirmed that both the baseline TMTV and pre-treatment dNLR were significantly associated with OS in the multivariate analysis [32]. Likewise, hyper-progressive disease (HPD), observed in one out of three patients (30%, *n* = 14/46) treated with ICIs, was more frequent in patients with a higher baseline TMTV, a higher number of metastatic sites, and pro-inflammatory parameters (pre-treatment dNLR and platelets counts) [32,33].

Pembrolizumab, as a single agent for first-line therapy for advanced NSCLC, is intended for use only in the patient’s group with PD-L1 TPS ≥ 50% [3,4]. However, some patients are primarily unresponsive to first-line pembrolizumab. While our score could be an easy and relevant strategy to detect the phenotype most likely to be associated with resistance to IT monotherapy, it may help to consider alternative therapies. Recently, the combination of platinum-based chemotherapy and immunotherapy has received FDA and EMA approval as a frontline therapy for advanced NSCLC patients, independent of PD-L1 status [34,35]. These developments raise the question on how and by which criteria the identification of the best treatment strategy will be carried out for the specific patient′s group with PD-L1 TPS ≥ 50%. In the era of precision medicine, identifying specific biomarkers that predict therapeutic effects from IO alone versus IO plus chemotherapy is a crucial step in guiding treatment selection. Efforts will consequently be made in our future studies to determine whether TMTV alone or in combination with dNLR (metabolic score) is associated with subgroups of patients who might benefit, or not, from IT plus chemotherapy, and to what extent these could improve clinical outcomes.

This study should be interpreted in the context of its main limitations. First, the retrospective nature of this study implies clinical or biological missing data. Unfortunately, the pre-treatment LDH levels were only available in 63% of the cohort, mainly because it was not systematically requested in each center at the time of the initial assessment. Consequently, the prognostic value of the lung immune prognostic index (LIPI), combining baseline dNLR and LDH [24,36], could not be investigated and balanced against our metabolic score. Second, while we validated the score in a retrospective manner, prospective validation in a larger, independent, and prospective cohort would provide the strongest assessment of the score. Third, PET images were acquired with four different devices, which may have any influence on the measurement of the PET features, but also indicates the generalizability of our model to different devices and centers. Finally, consensus guideline, iRECIST [37], which was developed by the RECIST working group for the use of modified RECIST version 1.1 in 2017, could not be considered as an endpoint, as it would have introduced a chronological bias. However, this did not alter our primary endpoint (OS) and recent studies demonstrated an excellent agreement between RECIST and iRECIST [38].

## 4. Materials and Methods

### 4.1. Patients Selection

In this retrospective study, patients with advanced NSCLC (stage IV or IIIB, which was ineligible for local therapy) who received first-line single-agent pembrolizumab between January 2017 and September 2019, and who had completed an FDG PET/CT scan at baseline, were included from four French centers. Of these, only those cases with PD-L1 expression TPS ≥ 50% and without ALK or EGFR aberrations were selected. The exclusion criteria were as follows: (i) patients with an FDG-PET/CT PET scan performed more than six weeks before the first treatment (*n* = 11), (ii) lost to follow-up patients (*n* = 3), and (iii) patients with advanced primary cancers other than NSCLC (*n* = 3). The workflow is provided in Figure 3. Medical records were reviewed for patient demographics, disease characteristics, blood and molecular parameters, treatment, and clinical outcome. The current study was conducted following the approval of the institutional review board of the Curie Institute, and informed consent was not required because of the retrospective character of the study.

### 4.2. FDG-PET/CT Acquisition

After a fasting time of at least 6 h, 18F-FDG PET/CT scans were performed 60 min (median 67 min; range, 53–84) after the injection of 18F-FDG (median activity 229 MBq; range,141–408). In most cases, images were obtained from the skull vertex to the proximal femur. The images were acquired and reconstructed according to current guidelines [39], using four PET/CT scanners (acquisition parameters for each device are provided in the Appendix A). Finally, the images were interpreted by two experienced physicians, board-certified in nuclear medicine (R.-D.S. and L.C.).

### 4.3. FDG-PET/CT Image and Biological Analyses

All of the FDG-avid metastatic lesions were selected for the analysis. As a measure of the tumor glycolytic activity, the 18F-FDG PET/CT uptake was quantified by maximum standardized uptake values normalized by body weight (SUVmax). To assess the metabolic tumor burden, MTV was measured by setting a margin threshold of 42% of SUVmax [19,40]. All of the values of SUVmax and MTV were automatically measured using the analysis software for each lesion. The tumor SUVmax was the maximum SUV of all of the lesions in a patient. TMTV was defined as the sum of the individual MTVs of all of the lesions analyzed, and the threshold of >75 cm^3^ was retained according to the metabolic score, as described previously [19,29].

Electronic medical records were reviewed for the pre-treatment of blood cell counts and LDH levels (within 28 days before the first pembrolizumab perfusion). For dNLR, a threshold of >75 cm^3^ was retained, according to the threshold from the largest published study with immune checkpoint inhibitors [24] and a metabolic score, as described previously [19,29]. For LDH and hemoglobin (Hb), we used the limit of each center (ULN—upper limit of normal) and the most commonly used threshold (anemia if Hb ≤ 12 g/dL), respectively.

### 4.4. Metabolic Score (TMTV and dNLR)

According to the metabolic scoring approach, the cohort was stratified into three groups, namely: a poor prognosis (dNLR > 3 and TMTV > 75 cm^3^), intermediate prognosis (dNLR > 3 or TMTV > 75 cm^3^), and good prognosis (dNLR ≤ 3 and TMTV ≤ 75 cm^3^) [19].

### 4.5. PD-L1 Expression (TPS)

The tumor PD-L1 expression was evaluated by using an immunohistochemistry analysis in the pretreatment biopsy samples, with a variety of PD-L1 immunohistochemical antibodies, namely: QR1 (*n* = 33), E1L3N (*n* = 24), SP263 (*n* = 3), and 22C3 (*n* = 3). According to previous studies [6], patients with a PD-L1 expression TPS between 50 and 89% were defined as “high PD-L1 expressors”. In contrast, patients with a TPS between 90 and 100% were defined as “very high PD-L1 expressors”.

### 4.6. Outcomes: Survival (OS and PFS) and Response Evaluation Criteria (DCR and ORR)

The clinical outcomes were the overall survival (OS), progression-free survival (PFS), disease control rate (DCR), and overall response rate (ORR)—definitions are provided in the Appendix A. The assessment of the outcome to be predicted was blinded. The responses were evaluated with a contrast-enhanced CT-scan, performed every 6 to 8 weeks, according to the RECIST1.1, which was the reference standard when the patients were evaluated.

### 4.7. Statistical Analyses

The continuous variables were dichotomized at their median value, except for those with the predefined thresholds discussed above. The Mann-Whitney-Wilcoxon test was used to compare the distribution of variables in the following sub-groups: patients with versus without corticosteroids, patients with high versus very high PD-L1 expression, and patients with squamous versus non-squamous NSCLC. Spearman′s rho coefficient was used to test the correlation between PET biomarkers. Survival curves were estimated for each group using the Kaplan-Meier method, and were compared statistically using the log rank test. The prognostic value of all of the factors was assessed with Cox models for survival. Multivariate analyses were performed using Cox proportional hazard regression models in a stepwise manner for independent significant factors. The factors associated with a clinical benefit (DCR and ORR) were tested with logistic regression. The Holm-Bonferroni correction was used to adjust for multiple comparisons (adjusted *p*). A *p* value of 0.05 or less was considered significant. Analyses were performed with PASW Statistics (version 25. SPSS Inc, Chicago, IL, USA) for Windows and R Studio.

## 5. Conclusions

As a conclusion, the metabolic score could provide clinicians an early indication of treatment failure in patients with PD-L1 TPS ≥ 50%, after the initiation of first-line pembrolizumab, using standard of care 18F-FDG PET scans and pre-treatment blood counts, which are both systematically performed in all advanced NSCLC patients before starting systemic therapies. The combination of baseline TMTV and pre-treatment dNLR identifies patient groups with markedly different prognoses, and offers a novel method for stratifying and selecting candidates who would not benefit from frontline anti-PD1. Future prospective research in a larger and independent cohort should determine how the score affects clinical decision-making, and how it can be used to guide treatment selection (IT alone or with chemotherapy) for advanced NSCLC in first-line setting.

## Figures and Tables

**Figure 1 cancers-12-02234-f001:**
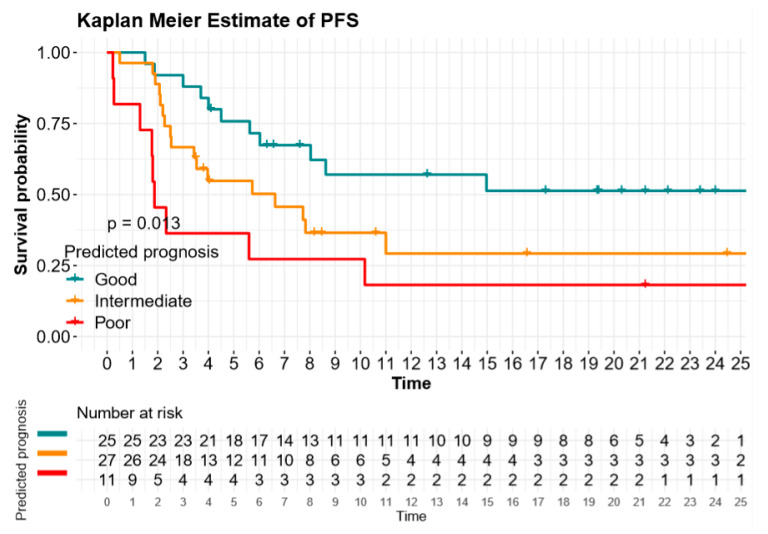
Progression-free survival (PFS) according to derived neutrophils-to-lymphocytes ratio (dNLR) and TMTV (metabolic score).

**Figure 2 cancers-12-02234-f002:**
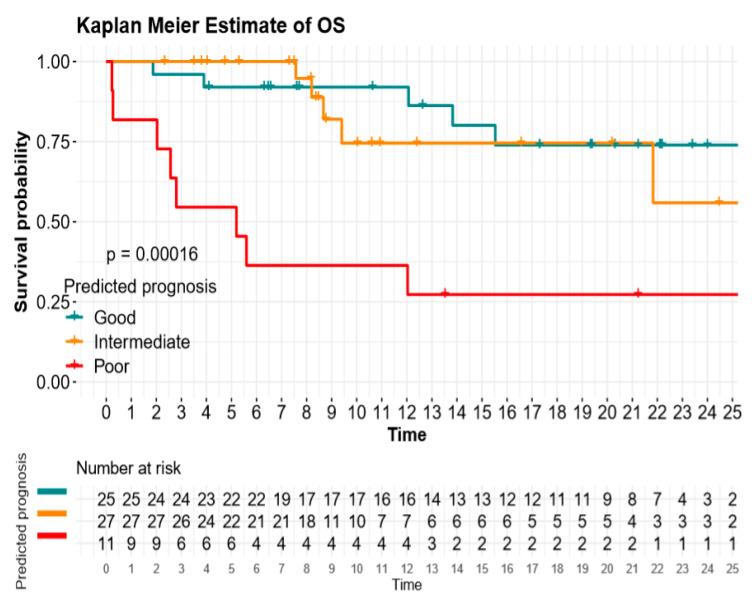
Overall survival (OS) according to dNLR and TMTV (metabolic score).

**Figure 3 cancers-12-02234-f003:**
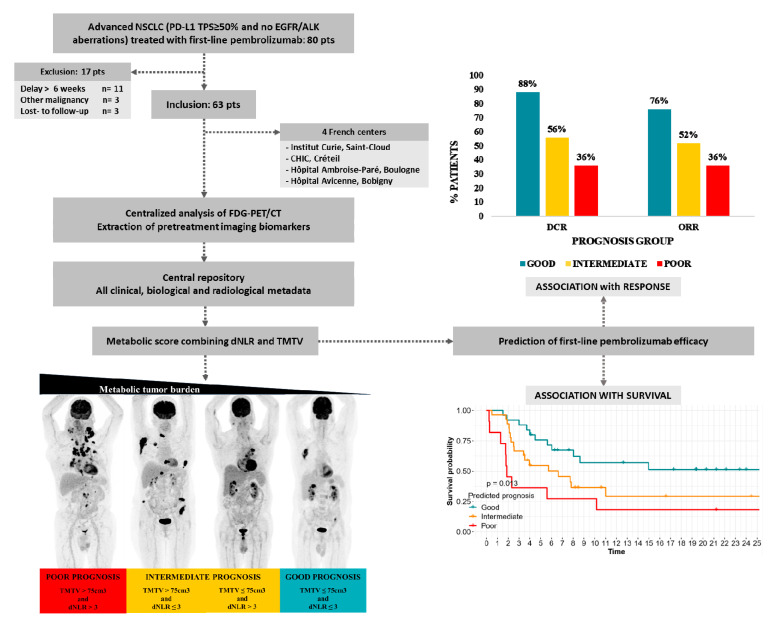
Work flow.

**Table 1 cancers-12-02234-t001:** Patient characteristics (*n* = 63).

Clinical Characteristics	Median (range), *n* (%)
*Patient demographics*	
Age (years)	65 (37–86)
Gender (Male/Female)	38 (60%)/25 (40%)
Body Mass Index (kg/m^2^)	24.0 (16.9–40.3)
Performance Status (ECOG)	1 (0–3)
Smoking history (current/former/no)	28 (44%)/32 (51%)/3 (5%)
*Histology*	
Non-squamous	50 (80%)
Squamous	13 (20%)
*PD-L1 expression*	
TPS: 90–100%	19 (30%)
TPS: 50–89%	44 (70%)
*Molecular alterations*	
BRAFv600E-ROS1-RET-METex14-HER2ex20	5 (8%)
KRAS	17 (27%)
Wild-Type	17 (27%)
Unknown	24 (38%)
*Biology*	
Neutrophils (G/L)	6.3 (2.4–15.8)
dNLR	2.2 (0.9–14,1)
Lymphocytes (G/L)	1.6 (0.3–3.9)
Hemoblobin (g/dL)	12.7 (8.3–16.0)
Platelets (G/L)	342 (134–639)
LDH* (UI/L)	242 (105–2197)
Albumin (g/dL)	35 (23–47)
C-Reactive-Protein (mg/dL)	26.4 (1.0–183.5)
*Treatment*	
Steroid use *	13 (21%)
*Staging*	
Stage (IIIb/IV)	10 (16%)/53 (84%)
Number of metastatic sites	2 (0–6)
Metastasis: adrenal/brain/liver/bone	17 (27%)/9 (14%)/9 (14%)/11 (17%)
**Pet Imaging Characteristics**	
*Tumor glucose uptake*	
Tumor SUVmax	18.0 (5.4–41.4)
*Metabolic tumor burden (TB)*	
Total metabolic tumor volume (TMTV) (cm^3^)	84.0 (12.4–427.9)
**Survival**	
Progression	37 (59%)
Death	18 (29%)
**Best Response Rate**	
Progressive disease	22 (35%)
Stable disease	4 (6%)
Complete response	2 (3%)
Partial response	35 (56%)

Note: * Available data for a total of 40 patients (63%).

**Table 2 cancers-12-02234-t002:** Univariate and multivariate Cox models for progression-free survival and overall survival.

	Progression-Free Survival				Overall Survival			
	Univariate		Multivariate		Univariate		Multivariate	
Variable	HR (CI 95%)	*p* Value	HR (CI 95%)	*p* Value	HR (CI 95%)	*p* Value	HR (CI 95%)	*p* Value
ECOG PS (≥2 vs. <2)	1.9 (0.9–4.0)	0.09	-	-	2.9 (1.0–8.6)	0.05	3.1 (0.9–9.6)	0.06
Histology (SCC vs. non-SCC)	1.1 (0.5–2.5)	0.74	-	-	1.1 (0.4–3.3)	0.77	-	-
Smokers (never vs. always)	0.4 (0.1–1.3)	0.12	-	-	1.4 (0.2–10.1)	0.78	-	-
PD-L1 expression (TPS ≥90 vs. <90%)	0.8 (0.4–1.7)	0.58	-	-	0.8 (0.3–2.4)	0.59	-	-
dNLR (>3 vs. ≤3)	2.2 (1.1–4.4)	0.02	2.0 (1.1–4.0)	0.04	3.6 (1.4–9.1)	<0.01	3.4 (1.3–8.8)	0.01
Hemoglobin (≤12 vs. >12 g/dL)	1.6 (0.8–3.2)	0.14	-	-	2.0 (0.8–4.9)	0.16	-	-
LDH* (>ULN vs. ≤ULN)	1.7 (0.8–4.0)	0.19	-	-	1.5 (0.5–5.1)	0.48	-	-
N metastatic sites (>3 vs. ≤3)	1.2 (0.5–2.7)	0.69	-	-	1.2 (0.4–3.5)	0.81	-	-
Liver metastasis (yes vs. no)	1.8 (0.8–4.2)	0.15			1.6 (0.5–4.7)	0.44	-	-
Tumor SUVmax (>16.5 vs. ≤16.5)	0.7 (0.3–1.3)	0.21	-	-	0.8 (0.3–2.0)	0.62	-	-
TMTV (>75 vs. ≤75 cm^3^)	2.0 (1.1–3.9)	0.04	1.8 (0.9–3.5)	0.08	2.9 (1.1–7.8)	0.03	2.1 (0.8–6.0)	0.09

Note: * Available data for a total of 40 patients (63%).

**Table 3 cancers-12-02234-t003:** Metabolic score and survival (PFS and OS).

Log-Rank Tests		Progression-Free Survival			Overall Survival		
Variable	*n*	Median PFS (Months)	95%CI	*p* Value	Median PFS (Months)	95%CI	*p* Value
Metabolic score				0.01			<0.01
Good (TMTV ≤ 75 cm^3^ and dNLR ≤ 3)	25	13.8	8.4–18.9		17.9	14.6–NR	
Intermediate (TMTV > 75 cm^3^ or dNLR > 3)	27	6.6	2.0–11.2		15.1	12.1–20.0	
Poor (TMTV > 75 cm^3^ and dNLR > 3)	11	1.9	1.3–2.5		5.2	1.9–8.5	

**Table 4 cancers-12-02234-t004:** Logistic regression of clinical benefit (disease control rate (DCR) and overall response rate (ORR)).

	DCR		ORR	
Parameter	*p* Value	OR (CI 95%)	*p* Value	OR (CI 95%)
ECOG PS				
≥2	0.34	1 (reference)	0.69	1 (reference)
<2		1.8 (0.5–6.3)		1.3 (0.4–4.9)
Histology				
SCC	0.76	1 (reference)	0.82	1 (reference)
Non-SCC		1.2 (0.3–4.3)		0.9 (0.2–3.0)
Smokers				
Never	0.85	1 (reference)	0.48	1 (reference)
Always		1.1 (0.1–10.5)		3.0 (0.3–9.5)
PD-L1 expression (TPS)				
90–100%		1 (reference)		1 (reference)
50–89%	0.83	0.9 (0.3–2.7)	0.93	1.0 (0.3–2.8)
dNLR				
>3	0.01	1 (reference)	0.04	1 (reference)
≤3		4.9 (1.5–16.3)		3.3 (1.1–10.5)
Hemoglobin				
≤12 g/dL	0.47	1 (reference)	0.62	1 (reference)
>12 g/dL		1.5 (0.5–4.4)		1.3 (0.5–3.7)
LDH *				
>ULN	0.07	1 (reference)	0.02	1 (reference)
≤ULN		2.8 (0.9–8.1)		5.8 (1.4–24.7)
N metastatic sites				
>3	0.14	1 (reference)	0.33	1 (reference)
≤3		2.7 (0.7–10.2)		1.9 (0.5–7.1)
Liver metastasis				
Yes	0.17	1 (reference)	0.35	1 (reference)
No		2.7 (0.6–9.4)		2.0 (0.5–8.2)
Tumor SUVmax				
>16.5	0.31	1 (reference)	0.24	1 (reference)
≤16.5		0.6 (0.2–1.6)		0.5 (0.2–1.5)
TMTV				
>75 cm^3^	0.02	1 (reference)	0.09	1 (reference)
≤75 cm^3^		3.8 (1.2–11.6)		2.5 (0.9–7.0)

Note: * Available data for a total of 40 patients (63%).

**Table 5 cancers-12-02234-t005:** Metabolic score combining TMTV and dNLR.

Logistic Regression	No Clinical Benefit			
	DCR		ORR	
	OR (95% CI)	*p* Value	OR (95% CI)	*p* Value
Metabolic Score (*n* = 63)				
Good (TMTV ≤ 75 cm^3^ and dNLR ≤ 3)	1 (reference)		1 (reference)	
Intermediate (TMTV > 75 cm^3^ or dNLR > 3)	5.6 (1.3–23.4)	0.02	2.7 (0.9–8.8)	0.08
Poor (TMTV > 75 cm^3^ and dNLR > 3)	9.8 (1.9–31.9)	<0.01	4.3 (1.0–18.3)	0.04

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
