# Peer review of "Association of the Metabolic Score Using Baseline FDG-PET/CT and dNLR with Immunotherapy Outcomes in Advanced NSCLC Patients Treated with First-Line Pembrolizumab"

_cancers, 2020, doi:10.3390/cancers12082234_

Round 1
Reviewer 1 Report
This is a retrospective study evaluating the association between a metabolic score characterized by Total Metabolic Tumor Volume evaluated by FDG PET and Neutrophils to Lymphocytes ratio and outcome in patients with NSCLC treated by first-line pembrolizumab. A total of 63 patients have been included.
The approach is not new being already used and proposed in a previous paper by the same author (Seban et al EJNM 2020) in a population of 80 patients with advanced NSCLC and treated with immunotherapy. On the other hand, compared to the previous study in the present study the prognostic meaning of the metabolic score has been investigated in a specific patient’s population with PDL1 expression > 50% and treated with immunotherapy as first-line treatment.
Furthermore, due to the retrospective nature of the study and the involvement of different centers some criticisms could derive from the use of different PET scans and immunostaining methods.
Nevertheless, the paper is easily readable, and the methodology used is correct.
1. Introduction
-See line 50 pag 2: “…only a fraction of patients experienced such therapeutic success…” Please report precise data in the text with references.
-Add the following reference: Polverari et al 18 F-FDG PET Parameters and Radiomics Features Analysis in Advanced Nsclc Treated with Immunotherapy as Predictors of Therapy Response and Survival. Cancers (Basel). 2020 May 5;12(5):1163. doi: 10.3390/cancers12051163Cancers 2020
2. Results:
Table 1: please add tumor stage (number of stage III and IV respectively). Best response rate includes progressive, stable, complete and partial response. Did the authors observe some cases of pseudoprogression in this cohort of patients and if yes how was the metabolic score in these cases? In accordance with previous studies results (ref 33,34)?
-Was there any patient with bone metastases? Only adrenal, brain and liver have been reported. Please comment and add the information if any.
-Correlation between biomarkers paragraph: The correlation between TMTV with age and LDH value are interesting findings but the correlation with the number of metastatic sites seems to me redundant being the TMTV extracted by the sum of all the lesions.
-Survival PFS and OS paragraph: In Table 2 please put the HR column before the p value column. P value in univariate analysis is most likely unadjusted in my opinion and adjusted in multivariate analysis.
3. Discussion: are outcomes in terms of overall response rate, PFS and OS of this present study consistent with results from previous clinical studies results focused on first-line immunotherapy in NSCLC (see Keynote 024 study for example)? And histological subtype (non-squamous vs squamous) may have an impact on study results? Please comments.
4. Methods: no major comments. See initial comments on different types of immunohistochemical staining and PET scans as intrinsic limitations of a retrospective study. The statistical analysis seems to be appropriate.
Minor comments:
Table 1: add % in Complete response line
Results, Survival PFS and OS paragraph: substitute “univariate and multivariate analysis” in “univariate and multivariate Cox models “
Reviewer 2 Report
The submitted article entitled “Association of the Metabolic Score using Baseline FDG-PET/CT and d NLR with Immunotherapy Outcomes in Advanced NSCLC Patients Treated with First-Line Pembrolizumab” provide an interesting contribution about “ an innovative way to select appropriate candidates who would not benefit from first-line 162 pembrolizumab monotherapy in advanced NSCLC patients” ( line 161-163)” . This is a well conducted study, supported on previous works released by the authors, and well discussed. The authors have been considered their limitations also. The use of neutrophil-to-lymphocyte ratio (NLR) is also interesting.
I kindly ask two questions to the authors:
- Could the authors provide some commentary about the reason to choose RECIST criteria better than EORT or PERCIST criteria to evaluate the clinical benefit in patients?
- What may be the cause because SUV max did not provide significant prognostic information or a more significant information?
Reviewer 3 Report
Based on previous separate investigations, this retrospective analysis combined to two predictive parameters, dNLR and TMTV, to rule-out patients that are unlikely to benefit from the anti-PD-1 therapy with Pembro, which is approved as first line treatment for NSCLC patients, with PD-L1 tumor proportion score (TPS) ≥50%. There are a few points for consideration and manuscript improvement as,
- Both dNLR and TMTV were tested previously and separately, it is assumed that the combination is better than either one used alone. Fig. 2 showed K-M estimates of PFS and OS with the combined parameters. Yet, Supplemental Fig. 2 only showed that with TMTV alone. The p-values are much better when combining the two vs using TMTV alone. However, the K-M results with dNLR alone is missing (might be previously published, would be OK to reproduced it in the Supplementary) for similar comparison.
- Why only combining dNLR with TMTV, since many parameters are not correlated? The Discussion acknowledged that many parameters were not available due to the retrospective nature of the study...
- How the ranges of the metabolic scores or their combinations were determined, e.g., TMTV >75 and dNLR>3 for the poor group? If each was determined separately, the optimal combination might be different! The regression results were calculated after these ranges were set, not before? Please explain or discuss.
Round 2
Reviewer 3 Report
I reviewed the revision. It is good to go.